# Vertical GaN MOSFET Power Devices

**DOI:** 10.3390/mi14101937

**Published:** 2023-10-16

**Authors:** Catherine Langpoklakpam, An-Chen Liu, Yi-Kai Hsiao, Chun-Hsiung Lin, Hao-Chung Kuo

**Affiliations:** 1Department of Photonics and Institute of Electro-Optical Engineering, College of Electrical and Computer Engineering, National Yang-Ming Chiao Tung University, Hsinchu 30010, Taiwan; cath01.ee09@nycu.edu.tw (C.L.); arsen.liou@gmail.com (A.-C.L.); 2Semiconductor Research Center, Hon Hai Research Institute, Taipei 11492, Taiwan; jason.yk.hsiao@foxconn.com; 3International College of Semiconductor Technology, National Yang-Ming Chiao Tung University, Hsinchu 30010, Taiwan; chun_lin@nycu.edu.tw

**Keywords:** GaN power device, MOSFET, breakdown voltage, specific on-resistance, electric field

## Abstract

Gallium nitride (GaN) possesses remarkable characteristics such as a wide bandgap, high critical electric field, robust antiradiation properties, and a high saturation velocity for high-power devices. These attributes position GaN as a pivotal material for the development of power devices. Among the various GaN-based devices, vertical GaN MOSFETs stand out for their numerous advantages over their silicon MOSFET counterparts. These advantages encompass high-power device applications. This review provides a concise overview of their significance and explores their distinctive architectures. Additionally, it delves into the advantages of vertical GaN MOSFETs and highlights their recent advancements. In conclusion, the review addresses methods to enhance the breakdown voltage of vertical GaN devices. This comprehensive perspective underscores the pivotal role of vertical GaN MOSFETs in the realm of power electronics and their continual progress.

## 1. Introduction

Power semiconductor devices serve as the fundamental building blocks within a power conversion system, exerting significant influence over factors such as system losses and switching speed. The performance of these power devices is primarily characterized by two key parameters: breakdown voltage (*V_BD_*) and on-resistance (*R_ON_*). These characteristics largely depend on the inherent material properties of semiconductors, including carrier mobility and critical electric field. Traditionally, silicon (Si) has been a cornerstone in the realm of power semiconductor devices. However, Si’s material limitations have constrained its further advancement. Consequently, there is an increasing demand for wide bandgap (WBG) semiconductors with bandgaps exceeding 3 eV and even ultra-wide bandgap semiconductors with bandgaps exceeding 4.5 eV [1]. Among the suite of wide bandgap materials, gallium nitride (GaN) has emerged as a pivotal material for crafting power devices owing to its exceptional material properties, including a wide bandgap of 3.4 eV, high critical electric field of 3.3 MV/cm, robust resistance to radiation, and impressive saturation velocity [2,3]. In addition to their high voltage handling capabilities, GaN-based power devices offer low *R_ON_* and minimal conduction losses due to GaN’s electron saturation rate, which is 2.8 times that of silicon [4,5]. Furthermore, GaN devices, especially GaN HEMTs, exhibit high-speed switching performance attributable to their small junction capacitance.

At present, GaN devices are categorized into three voltage ranges [6]. (a) Low (mid)-voltage devices: These devices have a maximum drain-to-source voltage (*V_DS, max_*) of less than 200 V. They find utility in various fields and are used in applications such as DC–DC power converters, motor drives, class D audio amplifiers, solar microinverters, synchronous rectifications, and LiDAR systems, among others. Depending on specific application requirements, these devices can exhibit *R_ON_* values ranging from approximately 2 mΩ to 200 mΩ [7,8,9,10]. (b) High-voltage devices: The *V_DS, max_* for these devices goes up to 650 V. They are used in applications such as industrial converters, telecommunication servers, servo motor control, power adapters, consumer electronics adapters, class D amplifiers, and data centers [11,12]. (c) Ultra-high-voltage devices: These devices have a *V_DS, max_* greater than 1 kV. Currently, there are few commercially available GaN power devices in the kV range. The highest voltage rating among commercially available GaN devices is approximately 900 V [13], which is used in applications like industrial applications, EV chargers, photovoltaic inverters, and more.

Leading companies such as Transphorm Inc., Infineon Technologies AG, GaN Systems Inc., Panasonic Corporation, Texas Instruments Inc., and others are actively working on introducing GaN devices with voltage ratings exceeding 1 kV. The distribution of WBG power devices based on voltage range is illustrated in Figure 1. Currently, several research articles demonstrate the feasibility of GaN power devices rated at 1 kV and above [14,15,16]. GaN power devices in the kV range will compete with other WBG power devices, including those based on SiC and Ga_2_O_3,_ in various applications, such as automotive, industrial, and photovoltaic sectors However, though some groups have reported achieving high blocking voltage GaN devices, the specific on-resistance (*R_ON-SP_*) of the GaN device is still much higher than that of the SiC counterpart [17,18].

## 2. GaN Power Devices

GaN-based power devices are primarily categorized into two fundamental structures: planar (lateral) structure devices and vertical structure devices, as depicted in Figure 2. Planar devices are typically manufactured on Si or SiC substrates, while vertical devices are homoepitaxially grown on GaN substrates. GaN HEMTs are the most commonly employed among GaN power devices. Si substrates are favored in their production to optimize cost-effectiveness and yield. However, epitaxially growing GaN on Si presents considerable challenges due to significant lattice mismatch and a substantial thermal expansion coefficient disparity between the two materials. The lattice mismatch can lead to dislocation propagation and defect generation throughout the GaN epitaxial layers, while thermal mismatch may cause cracking in the GaN layer during cooling after epitaxial growth [20,21,22]. To mitigate defect propagation and attain high-quality, defect-free GaN on Si, a meticulous optimization of the buffer region is imperative. Additionally, it is essential to limit tensile stress during both the growth and cooling phases of the process [23,24].

In addition to challenges related to epitaxial layer mismatch, the quality of the top surface becomes a critical concern for achieving optimal performance in lateral GaN devices. The presence of surface states on the AlGaN layer significantly influences the formation of the two-dimensional electron gas (2DEG) [25]. These states can have adverse effects on device performance by unintentionally depleting the 2DEG. This depletion can create a virtual gate between the gate and drain regions, leading to degradation in device performance and reliability. To counteract the depletion of the 2DEG, a well-optimized silicon nitride (SiN) passivation layer is typically employed.

Currently, commercially available GaN HEMTs offer a *V_BD_* of approximately 650/900 V, coupled with a maximum output DC rating of 150 A [26]. These devices find applications across various fields, including onboard battery chargers, high-efficiency power converters, and dense solar panel inverters. However, lateral GaN devices face specific limitations when it comes to scaling *V_BD_* and ensuring reliability. These limitations are primarily associated with lateral carrier flow from the source to the drain, with the electron density in the channel being highly sensitive to surface traps and buffer defects. Furthermore, these imperfections in the device can lead to phenomena such as current collapse and dynamic degradation of *R_ON_* [26,27,28]. The severity of these adverse effects increases as *V_BD_* ratings rise. Moreover, the lateral electron transport nature results in a nonuniform distribution of the electric field within the device, leading to a higher electric field concentration in specific regions such as the gate, field plate, or drain edges. This nonuniform electric field distribution may result in premature semiconductor and dielectric breakdown, along with increased electron trapping at the surface. These factors contribute to performance degradation and limit the device’s voltage-blocking potential [28,29].

Furthermore, the *V_BD_* of lateral devices is directly proportional to the distance between the gate and drain, which increases both the device size and cost. Another significant concern with lateral GaN HEMT devices is that they are typically normally-ON devices. While various methods exist to create normally-OFF devices, such as using fluorine ion implantation [30,31,32], recess gates [33], p-GaN gates [34], trigate structures [35,36], cascode configurations [37,38], etc., the threshold voltage (*V_TH_*) achieved is typically around 1–2 V in many cases, including commercial devices. This characteristic makes the device less than ideal for failsafe operation. Lastly, the absence of avalanche capability in lateral GaN devices, possibly due to the lack of a p–n junction between the drain and source regions, which aids in hole extraction during impact ionization, poses challenges in preventing the device from short-term overvoltage failure [39].

To address the limitations of lateral GaN devices, there has been a growing exploration and investigation of vertical GaN devices. The primary distinction between vertical and lateral devices lies in the direction of current flow, as illustrated in Figure 2 with a red arrow for both types. In a vertical device, current flows vertically from the bottom to the top surface (with electrons carrying the current from top to bottom) or in parallel with the direction of the GaN epitaxial layers. Consequently, current conduction occurs through the surface channel and then through a bulk drain drift region formed by a homoepitaxial GaN layer on a GaN substrate. Unlike lateral devices, vertical devices do not have a 2DEG drain drift region near the surface or a defect-rich buffer layer, as found in GaN HEMTs. As a result, there is less degradation in dynamic *R_ON_*, which often occurs due to trapped impurity charges or bulk traps resulting from lattice mismatch [40]. One of the most significant advantages of vertical devices over lateral ones is their *V_BD_*. The *V_BD_* of a vertical device can be increased by augmenting the thickness of the N-drift layer, which is typically a low-doped or unintentionally doped GaN epitaxial layer (also known as the GaN N-drift layer). Consequently, this makes the *V_BD_* independent of the lateral size of the device. Moreover, a vertical GaN device can achieve a *V_TH_* ranging from around 5 to 15 V, making it well suited for high-power applications [41].

Another advantage of vertical GaN devices is their ability to exhibit avalanche breakdown, enhancing device reliability and potentially eliminating the need for overdesigning the device [42,43,44]. During transients, avalanche breakdown serves as a failsafe operation, allowing the device to operate closer to its material limits [45]. The *V_BD_* of a vertical GaN device can be significantly higher because the drain is located at the bottom of the device. When a high voltage is applied at the terminal, the electric field is evenly distributed throughout the device in a vertical direction, unlike lateral devices that experience electric field spikes at the gate edge [46]. This even distribution of the electric field within the device, away from the surface, can mitigate the effects of current collapse by countering the influence of surface states and thus slowing down the occurrence of current collapse [47,48]. Vertical GaN devices also offer the advantage of easily increasing power density. Since there are no electric field spikes, unlike lateral devices, there is no need for the field plate structure, which typically increases gate leakage spacing, to enhance breakdown performance. This makes it simpler to improve wafer utilization and boost power density [49].

The nFET operation of the vertical device is controlled by two primary p–n junctions, forming gate–source and gate–drain diodes. Avalanche breakdown occurs when the drain–source voltage of the device exceeds its *V_BD_*. Initially, avalanche breakdown occurs through the reverse-biased drain–gate diode region, leading to an increase in the gate–source voltage and the opening of the channel region. Unlike lateral GaN devices, avalanche breakdown in vertical devices remains within the short-term power dissipation limits. Furthermore, energy dissipation occurs within the drift region of the device rather than on the surface (a sensitive region). This feature helps protect the device from transient spikes and other abnormal operating conditions. In a vertical GaN device, the epitaxial layer is grown homoepitaxially on the native GaN substrate, eliminating issues related to lattice or thermal expansion coefficient mismatches. The performance of the vertical GaN device remains unaffected by defect densities and manufacturing challenges such as wafer bow, warp, cracking, etc. Consequently, vertical GaN devices offer high reliability, yield, and, ultimately, reduced costs compared to lateral devices. Despite the significant potential of vertical GaN in dominating power devices, the current landscape still heavily relies on lateral device structures due to the complexities involved in preparing vertical devices and the high cost of GaN substrates [50].

A notable breakthrough in GaN devices is the introduction of GaN on SiC. GaN on SiC is a device that combines the advantages of both materials. The remarkable thermal conductivity of SiC, outperforming GaN alone, facilitates efficient heat dissipation, fostering enhanced device performance and durability. These devices are characterized by high power density and high-frequency response, attributes that lend themselves perfectly to applications in high-frequency electronics like radars and satellite communications. The symbiotic relationship between GaN and SiC, marked by minimal lattice mismatch and thermal stress, ensures devices that are reliable over extended operational timelines. This integration is known to reduce parasitic capacitances and resistances, paving the way for quicker device switching and diminished energy losses [51,52].

Despite the obvious advantages, SiC substrates tend to be more expensive compared to alternatives like Si or sapphire, which can escalate the overall device cost. The nuanced processes involved in the epitaxial growth of GaN on SiC lead to defects, which could compromise device efficiency. The demand for large-diameter SiC wafers often outstrips the supply, potentially hindering widespread commercial adoption and mass production [53,54]. A major issue with these devices is their compromised off-state performance, characterized by increased leakage and reduced *V_BD_*. Studies suggest that these defects can intensify leakage by acting as electron–hole pair generators under strong electric fields. As illustrated in some research, when exposed to high electric fields, these defects create holes that can accumulate at the boundary between the dielectric and the semiconductor. This accumulation can then intensify the field in the oxide, leading to a potential device failure. Interestingly, in comparisons using trench MOSFETs, devices on bulk GaN outperform those on alternative materials [55,56]. GaN on SiC in vertical devices is a tapestry of clear advantages punctuated by challenges.

## 3. Vertical GaN Field Effect Transistors

Several types of vertical GaN devices have been developed, including p–i–n diodes [57,58], vertical GaN Schottky barrier diodes (VSBDs), and vertical GaN transistors. Among vertical GaN transistors, two primary types exist: (a) current aperture vertical electron transistor (CAVET) and (b) vertical GaN MOSFET. Figure 3a–d illustrate the schematic structure of vertical GaN devices. Vertical GaN MOSFETs can be further categorized into two subtypes: (a) planar GaN MOSFET and (b) trench GaN MOSFET. Planar GaN vertical MOSFETs are crafted using ion implantation to create regions like p-well and n+ source contact regions, resulting in a relatively cost-effective fabrication process [59]. However, significant challenges include achieving precise control of the acceptor concentration in the p-well to determine the *V_TH_* and addressing the formation of residual point defects during p-well ion implantation and post-implantation annealing. These factors are crucial for ensuring stability in the *V_TH_* and overall device reliability [60]. As a result, from the early stages, the trench MOS structure of the gate is adopted since the p-well can be formed by epitaxial growth to avoid the immature ion-implantation of p-GaN in device fabrication.

The concept of the CAVET structure was introduced in 2004 [61]. This device structure bears similarity to a double-diffused metal-oxide-semiconductor (DDMOS) structure and exhibits a negative VTH due to the presence of a channel region beneath the gate, arising from the AlGaN/GaN heterointerface. However, the vertical trench GaN MOSFET is a normally-off device, which is preferred for safe power device operation. Therefore, trench GaN MOSFETs have gained preference over CAVET devices. Moreover, they offer a relatively straightforward manufacturing process, eliminating the need for regrowth of the AlGaN/GaN layer [62,63].

In 2007, ROHM Co. Ltd. (Kyoto, Japan) achieved a significant milestone by reporting the first-ever GaN trench MOSFET [64]. The following year, in 2008, a normally-off vertical GaN trench MOSFET, utilizing a common MOS structure, was introduced with an impressive *V_TH_* of approximately 3 V [65]. Fast-forwarding to 2014, T. Oka et al. achieved a breakthrough by presenting a vertical GaN trench MOSFET on a freestanding GaN substrate. This device boasted a formidable *V_BD_* of around 1.6 kV, along with a 7 V *V_TH_*. It employed a field plate termination strategy around the mesa isolation, effectively reducing electric field crowding at the p–n junction edge [41]. Then, in 2015, T. Oka et al. extended their pioneering work to unveil a vertical trench MOSFET featuring a low *R_ON-SP_* of 1.8 mΩ·cm^2^. This remarkable device had a *V_BD_* of approximately 1.2 kV and operated at over 20 A [66]. This modified device was based on the authors’ prior work [41] and featured a hexagonal trench gate layout, effectively reducing the *R_ON-SP_* by increasing the gate width per unit area.

In 2016 [67], the same research group made notable advancements. They demonstrated a single-cell vertical GaN MOSFET featuring an impressive *V_BD_* of around 1.6 kV and *R_ON-SP_* of approximately 2.7 mΩ·cm^2^ for a single cell. These achievements were realized by decreasing the n-GaN drift layer doping concentration of the previous work [66]. Notably, the group also reported the fabrication of a multicell MOSFET with a chip size measuring 1.5 mm × 1.5 mm, which exhibited a *V_BD_* of roughly 1.3 kV. This device could operate at a drain current of up to 23.2 A and demonstrated rapid switching characteristics. Moving on to 2016, a normally-off 1.2 kV “all-gate” vertical MOS-gate transistor was designed and successfully fabricated [68]. This innovative structure featured two MOS gates on both sides of the source pillar and achieved an *R_ON-SP_* of approximately 2.8 mΩ·cm^2^. In 2020 [59], researchers introduced a 1.2 kV vertical GaN planar MOSFET with an effective on-resistance of 1.4 mΩ·cm^2^, employing an ion implantation process. To enhance the device’s *V_BD_*, they executed a sequential implantation process using Mg and N. Additionally, a short cell pitch design was implemented to improve the device’s *R_ON_*.

In the most recent developments of 2022 [69], a 1.3 kV vertical GaN MOSFET was introduced on a 4-inch freestanding GaN substrate. This device featured a *V_TH_* of approximately 3.15 V and demonstrated a low *R_ON-SP_* of 1.93 mΩ·cm^2^. These achievements translated into an impressive power figure of merit (PFOM) of 0.88 GW/cm^2^. Figure 4 provides a comparison of the PFOM for vertical GaN MOSFETs with breakdown voltages of 1 kV and above, highlighting the progress made in this field over the years.

While GaN MOSFETs hold great promise for high-performance power devices, they encounter two significant challenges during fabrication and operation. Firstly, the device exhibits low mobility within the p-GaN inversion layer, necessitating higher gate voltages for device activation. Secondly, achieving reliable ohmic contacts becomes problematic after the plasma etching process in the buried p-type region (used for body contact) as the generation of n-type compensating vacancies increases during the etching process [70]. Moreover, the p-type GaN layer has lower acceptor activation and much lower mobility compared to the n-type. To overcome these challenges, innovative GaN transistor structures have been introduced, including fin structures [71,72] and in situ oxide–GaN interlayer FETs (OG-FETs) [73,74]. Figure 5 illustrates schematic diagrams of fin structure GaN transistors and OG-FETs.

The novel GaN vertical fin power field-effect transistor features a fin-shaped n-GaN channel enclosed by gate oxide and a gate electrode, enabling precise control of current flow. This device exclusively relies on n-type GaN and eliminates the need for epitaxial growth. The device achieves normally-off operation by ensuring the fin width remains below 500 nm. The difference in work function between the gate electrode and GaN allows for complete electron depletion in the channel, resulting in normally-off operation without requiring a p-GaN layer. In [72], a 200 nm fin width normally-off vertical GaN FinFET with a 1.2 kV *V_BD_* and a 0.8 V *V_TH_* was reported. This device exhibited a low *R_ON-SP_* of approximately 0.2 mΩ·cm^2^, translating to an impressive PFOM value of 7.2 GW/cm^2^. It was noted that a larger device could achieve a higher current of around 10 A at 800 V. However, both large and small devices faced the challenge of catastrophic breakdown, highlighting the impact of the absence of the p-GaN layer on device robustness. Additionally, strict control over fin width to achieve a normally-off device increased fabrication costs [71].

On the other hand, the OG-FET trench MOSFET structure enhances channel mobility without compromising its normally-off characteristics. Generally, in an MOSFET, oxide growth occurs ex situ using atomic layer deposition (ALD) after trench etching. In contrast, OG-FET leverages MOCVD for a regrowth process [75]. The improved channel mobility can be attributed to the in situ oxide and the regrowth of a thin unintentionally doped (UID) channel layer. The UID GaN channel improves the channel mobility by reducing impurity scattering. In [73], a 1.4 kV GaN OG-FET with a double field-plated structure was reported. This device had *V_TH_* of 4.7 V and *R_ON_* of approximately 2.2 mΩ·cm^2^.

In the following discussions, we will list the process skills to improve the VBD through electric field management in various types of vertical GaN MOSFET power devices. Finally, we will summarize the vertical GaN device performance and then benchmark it against lateral GaN MOSFETs and SiC power devices.

## 4. Techniques to Boost the Breakdown Voltage

The most prevalent approaches to enhance the *V_BD_* in GaN vertical devices, including GaN vertical MOSFETs, typically involve the following methods: (a) lowering n-GaN drift layer doping concentration, (b) optimizing n-GaN drift layer thickness, (c) incorporating field plates, and (d) implementing edge termination techniques. Furthermore, the *V_BD_* of GaN vertical MOSFETs is also influenced by the characteristics of the metal-oxide-semiconductor capacitor (MOSCAP) within the device. The thickness of the gate oxide affects the distribution of the electric field, as the gate oxide has a higher ability to withstand a higher electric field, thus increasing the breakdown voltage, while a thinner gate oxide increases the leakage current through the gate, resulting in premature breakdown. However, increasing the gate oxide thickness increases MOSCAP, thus reducing the speed of the device [76].

Moreover, the durability of the oxide dielectric, particularly a high-quality gate oxide, is pivotal as it can withstand a substantial electric field, ultimately contributing to an augmentation in the *V_BD_*. The dielectric constant (*k*) of the gate oxide plays a crucial role in determining the device’s performance. A high *k* gate oxide can achieve a thicker gate dielectric with the same capacitance as that of a smaller *k*, thus achieving a higher *V_BD_* with a similar speed as that of a smaller *k*. Additionally, a higher *k* gate oxide reduces the gate leakage current, thus improving the *V_BD_* [77]. However, although high *k* gate oxide can improve *V_BD_* and reduce gate leakage, high *k* gate dielectric materials are susceptible to thermal stress; hence, proper thermal management is required to avoid reliability issues. The gate dielectric material with positive valence band offset with GaN, namely, SiO_2_, Al_2_O_3_, etc., tends to exhibit catastrophic breakdown, while the dielectric with a negative valence band offset, namely, SiN, shows an increase in offset leakage current [78].

### 4.1. Drift Layer Thickness and Doping Concentration

Although the GaN vertical MOSFET is grown homoepitaxially on a bulk GaN substrate, the amount of dislocation density present is still very high. To achieve a high-performance GaN device, an improved epitaxial layer is required. Additionally, the thickness and doping concentration of the buffer or the drift layer also play important roles in enhancing the device’s performance [79]. A thicker drift or buffer GaN layer improves the material quality by reducing defect density and resulting in better surface morphology. Moreover, a thicker drift layer can filter dislocations during epitaxial growth and prevent dislocations from propagating towards the upper layers [80,81,82].

Another critical factor for enhancing performance is the optimization of doping concentration in the drift region. To obtain the optimized doping concentration for the drift region, it is imperative to understand the dependence of doping concentration on the *V_BD_* and the *R_ON-SP_* of the device. One of the most preferred benchmarks for power devices is the *PFOM* (power figure of merit) given by [83,84,85]:(1)$$PFOM=\frac{{V_{BD}}^{2}}{R_{ON-SP}}$$

The connection between the doping concentration and the *R_ON-SP_* of the drift region can be assessed through majority carrier conduction, as expressed by the following equation:(2)$$R_{ON-SP}=\left(\frac{t_{Drift}}{qn\mu_{n}(n)}\right),$$
where *t_Drift_* represents the thickness of the drift layer, *n* signifies the doping concentration, and *μ_n_* denotes the electron mobility, which depends on the doping concentration. The correlation between electron mobility and doping concentration is elucidated by the Mnatsakanov empirical mobility model [86], given by
(3)$$\mu_{N}\left(n\right)\mu_{min}+\left(\frac{\mu_{max}-\mu_{min}}{{1+(n/N_{G})}^{\gamma }}\right),$$

In this equation, *μ_min_*, *μ_max_*, *N_G_*_,_ and *γ* are the parameters used for fitting. However, establishing a direct relationship between the doping concentration of the drift layer and the *V_BD_* is not a straightforward task. It necessitates a thorough examination of the primary causes of the avalanche effect, which can be attributed to either parallel-plane breakdown or punch-through breakdown [87]. Parallel-plane breakdown, also referred to as non-punch-through breakdown, primarily depends on factors such as the doping concentration of the drift region and the impact ionization coefficient of the material rather than the thickness of the drift layer. The distribution of the electric field within the device for both types of breakdowns is illustrated in Figure 6:

In the case of parallel-plane breakdown, the electric field distribution takes on a well-defined triangular shape across the drift region. Under these conditions, determining the maximum electric field is a straightforward task, and device breakdown occurs when this maximum electric field reaches the critical electric field (*E_crit,NPT_*) characteristic of the device. The typical critical electric field (*E_crit_*) in GaN can be computed using the following equation, which draws upon values reported by Maeda et al. [89]:(4)$$E_{Crit}=\left(\frac{2.162\times 10^{6}+800\times T}{1-0.25.\log_{10}\left(n/10^{6}\right)}\right),$$
where *T* is the temperature in kelvin. In the scenario of punch-through breakdown, the electric field distribution within the drift region assumes a trapezoidal shape, primarily because the drift region becomes entirely depleted of charge carriers. Within this configuration, the electric field experiences a more gradual change in the drift region, owing to its lower doping concentration, while it exhibits a rapid variation over distance in the n+ doping region due to the higher doping concentration [90]. The breakdown voltage under punch-through conditions is given by [85]
(5)$$V_{BD\_PT}=V_{BD\_PP}\left(\frac{d_{Drift}}{W_{pp}}\right)\left(2-\frac{d_{Drift}}{W_{pp}}\right),$$
where *V_BD_PT_* is the breakdown due to punch-through, *V_BD_PP_* is the parallel-plane breakdown voltage, *W_pp_* is the parallel-plane depletion region, and drift is the drift layer thickness. *V_BD_PP_* is given by *V_BD_PP_ =* (*E_crit_.W_pp_/*2) and *W_pp_ =* (*ɛ_s_EC_rit_/eN_d_*). The validity of the previously mentioned equation for the breakdown voltage of the punch-through region holds when the critical electric field for punch-through (*E_crit,PT_*) closely approximates the *E_crit,NPT_*, meaning that *E_Crit,NPT_ ≈ E_Crit,PT_ ≈ E_Crit_*. According to findings reported in [89], the disparity between the critical electric fields for parallel-plane and punch-through breakdown is exceedingly minimal.

In the punch-through state, the *V_BD_* is influenced not only by the doping concentration but also by the thickness of the drift region. It is worth noting that the typical breakdown voltages for punch-through and parallel-plane conditions can significantly differ under certain doping conditions [91], as shown in Figure 7. To obtain the optimum drift doping concentration, the relation between the PFOM and the doping concentration shown in Figure 7 should be considered. The region where PFOM has the highest value is considered to be the optimized drift doping concentration.

### 4.2. Field Plate Termination

Another widely adopted technique used to enhance the *V_BD_* of power devices is the incorporation of field plate (FP) technology. GaN-based HEMTs have particularly benefitted from this approach, as it helps increase the *V_BD_* by mitigating electric field concentration at the edges of the gate or drain electrodes [92,93,94]. During the off state of the power device, the electric field distribution is not uniform across the device. Instead, there are areas of high electric field concentration, specifically located at the edges of the gate or drain electrodes. This heightened electric field can lead to increased leakage current in these regions, attributed to the tunneling effect, thereby causing premature breakdown of the device. Moreover, the presence of this high electric field can impact the charging and discharging processes of the device, potentially resulting in switching delays.

In the vertical GaN MOSFET, there is a highly doped n-type GaN region for the ohmic contact and a lightly doped n-type GaN drift layer for voltage blocking. Due to these structural distinctions, the distribution of electric fields within the device differs from that in lateral GaN devices. Electric field crowding primarily occurs at the interface between the gate metal and the semiconductor. This leads to the formation of a depletion region around the edge of the gate, where the electric field resulting from the space charge accumulation intensifies, significantly enhancing the electric field strength in that region [95]. Figure 8 depicts the schematic structure of a vertical GaN OG-FET with a field plate [73]. Through the use of the first field plate (FP1), the high electric field at the gate metal edge is reduced from 3.7 MV/cm to 2.6 MV/cm [96]. The field enhancement factor, *η,* which characterizes the degree of field crowding, is used to investigate the effect of field crowding quantitatively and is given by
(6)$$\eta =\frac{E_{M}}{E_{0}},$$
where *E_M_* represents the maximum gate-edge electric field, and *E*_0_ denotes the electric field at the opposite end of the gate [95]. In the case of the device without FP, the field enhancement factor exhibits a linear dependence on the depth of the depletion region. As the depletion region becomes deeper, there is a greater concentration of the electric field stemming from the nearby space charge near the gate edge, resulting in an augmented field enhancement factor, *η* [97]. An empirical formula of *η* and the depletion region depth (*W_DD_*) is given by [95]
(7)$$\eta =1+aW_{DD},$$
where the parameter “*a*” is approximately 1.5 μm^−1^ and exhibits some degree of dependency on various structural parameters. The relationship of *η* on *W_DD_* is different for different types of breakdown. For parallel-plane breakdown, *η* is the same as the formula given in Equation (7). For punch-through condition, *η* is given by
(8)$$\eta =1+aT_{GaN}$$
where *T_GaN_* is GaN drift layer thickness. With FP, the depletion region extends beyond the gate edge, partially overlapping with the FP electrode. Consequently, the concentration of the electric field at the gate edge disperses towards the edge of the FP electrode. This redistribution results in a reduction of the maximum electric field at the contact edge, thereby decreasing η. While the incorporation of the FP improves the device’s *V_BD_*, it also introduces another high-field region situated within the semiconductor and the dielectric material beneath the FP edge region, and the breakdown in this region becomes another issue to be concerned. The effect of FP also depends on the type of dielectric materials as well as the height of the FP from the semiconductor.

### 4.3. Edge Termination

An established technique for improving *V_BD_* entails employing an edge termination method to mitigate the concentration of electric field lines near the edge of the p–n junction. This technique is closely linked to the phenomenon of impact ionization, which exerts a significant influence on the *V_BD_* of the device [98]. Edge termination represents a methodology for modifying the contour of the depletion layer at the junction’s edge. This can be achieved through various means, including etching, anode extension, ion implantation, and dielectric passivation. In the realm of GaN devices, various types of edge termination techniques have been reported, such as mesa etches, floating guard rings, plasma treatment, junction termination extension [99], ion implantation [57], and bevel termination [100]. The application of edge termination techniques does not compromise the rectifying properties of the device. Among these techniques, ion implantation stands out as a widely adopted method in Si and SiC power devices. However, in the context of GaN devices, ion implantation is still relatively less utilized, as indicated in reference [101]. Ion implantation is employed in GaN devices to achieve isolation by introducing material damage through controlled bombardment, resulting in the creation of mid-gap defects [102,103,104], hence, creating a highly resistive edge that facilitates the lateral distribution of the electric field.

Another technique that utilizes ion implantation involves managing the charge on the edge termination by offsetting the region through the introduction of counter-doping in p-GaN with Si, as detailed in reference [105]. Additionally, N ion implantation can be employed to create a semi-insulating region by controlling the total dose within the edge termination [104]. The ion implantation method is also employed in techniques such as junction termination extension (JTE) and field-limiting rings (FLR). In the case of JTE, an ion implantation process is used to create a thin, lightly doped layer encircling the diffusion junction area. Figure 9 illustrates a device featuring FLR edge termination, where the extension of this thin layer results in the expansion of the depletion layer, akin to the effect of a field plate [106]. This extension helps to reduce the issue of electric field crowding at the junction. Nevertheless, a major challenge faced by GaN devices lies in the difficulty of establishing p-type regions through ion implantation, as highlighted in reference [107].

Mesa termination stands out as the most widely adopted technique in vertical GaN power rectifying devices. This method involves selectively removing a portion of the device using dry etching techniques. Typically, the use of a 90° angle mesa termination is avoided because it leads to significant electric field crowding, which can reduce the device’s VBD. However, a study conducted by Fukushima et al. [108,109] reveals that deep mesa termination can enhance the *V_BD_* and reduce reverse leakage. The device with deep mesa (5–10 µm) exhibits a uniform electric field distribution across the entire device, while shallow mesa (1–2 µm) tends to have increased leakage and lower *V_BD_*. Another commonly employed mesa termination technique for GaN devices is the beveled mesa, where the heavily doped region is removed to a greater extent than the lightly doped side [90]. Figure 10 illustrates a schematic representation of mesa edge termination in GaN power devices [96]. Furthermore, to mitigate electric field crowding at the edge, a combination of field plates with mesa termination is often used. Mesa termination, when coupled with a field plate, can alleviate the electric field at the junction without the need for ion implantation [45,110]. However, it is worth noting that this intricate process can increase the fabrication costs of the device.

The choice of the optimal method for improving *V_BD_* depends on specific requirements, necessitating careful consideration of associated trade-offs. The advantages and trade-offs of the abovementioned methodologies are as follows. (1) Modifying doping concentration in the drift region: decreasing doping concentration increases *V_BD_*; however, *R_ON_* increases. Conversely, increasing doping concentration improves *R_ON_* but reduces *V_BD_*. (2) Drift region scaling: increasing drift thickness enhances *V_BD_* but simultaneously increases device *R_ON_*, thus reducing the switching speed of the device. (3) Gate oxide engineering: optimizing gate oxide thickness and material properties can influence GaN MOSFETs’ voltage-blocking capabilities. Thicker gate oxides enhance *V_BD_* but elevate gate capacitance, impacting overall performance. (4) Field plate termination: field plate termination is a widely adopted method for improving *V_BD_*. It facilitates even distribution of the electric field, reducing electric field crowding at the gate-edge region, thus improving voltage handling capabilities and device reliability. However, the efficacy of field plate termination diminishes as device size decreases. (5) Edge termination techniques: edge termination techniques aim to enhance *V_BD_* at the device’s edges, especially beneficial for high-voltage applications. These techniques effectively mitigate the risk of edge breakdown, offering simplicity and scalability. Nevertheless, the dry-etched sidewalls in edge termination introduce a leakage path that significantly impacts the off-state drain current [111]. Ion implantation has demonstrated significant improvements in Si and SiC devices by protecting the dielectric at the trench bottom and serving as a body contact. In GaN, limitations exist for p-ion implantation and selective area doping. An alternative approach involves creating a buried shield under the trench using etching and regrowth methods. Ion implantation has shown significant improvement in Si and SiC-based devices to protect the dielectric at the bottom of the trench and to serve as body contact. However, due to the limitation in p-ion implantation and selective area doping in GaN [112], a buried shield is formed under the trench by the etching and regrowth method [113].

## 5. GaN Vertical Transistor Summary and Benchmarking

Figure 11 presents a schematic comparison of GaN vertical MOSFETs designed for *V_BD_* exceeding 1 kV. To achieve a high *V_BD_*, these devices employ both FP termination and mesa termination to eliminate potential crowding issues at the p–n junction edge. To reduce *R_ON_*, modifications are made to the thickness and doping concentration of the channel and drift layers. Reducing the doping concentration and thickness of the p-GaN layer decreases channel resistance, while adjusting the thickness and increasing the doping concentration of the n-GaN drift layer reduces drift resistance, thus increasing the *V_BD_*. Additionally, further *R_ON_* reduction can be achieved by increasing the gate width per unit, which in turn increases current density. This design improvement resulted in a reduction in *R_ON-SP_* from 12.1 mΩ-cm^2^ [41] to 18 mΩ-cm^2^ [66]. To increase the *V_BD_* from 1.2 kV [66] to 1.6 kV [67], the doping concentration of the n-GaN drift layer is reduced. However, there is a tradeoff between *V_BD_* and device switching speed. As *V_BD_* increases, *R_ON_* also increases, which can slow down device performance. In addition, reducing leakage current can enhance the *V_BD_* [59]. Sequential N implantation after Mg implantation significantly reduces leakage current by suppressing hole traps, resulting in improved *V_BD_* [59,114]. In addition, enhancing the MOS interface quality and stability can also improve breakdown characteristics. A two-step process introduced in [69] involves thorough surface cleaning followed by (NH_4_)_2_S passivation to improve the MOS interface. Furthermore, the lower electron mobility of p-GaN increases channel resistance, requiring a higher gate bias to lower resistance [115]. Therefore, a GaN interlayer is grown between the in situ oxide layer and the trench to increase channel mobility.

Table 1 presents a comparison of different vertical GaN transistors featuring *V_BD_* exceeding 1 kV. Generally, larger devices exhibit higher *V_BD_*, but this comes at the cost of increased *R_ON-SP_*. In the context of power devices, it is crucial to evaluate the PFOM, where higher values signify superior voltage-blocking capabilities. Although elevated *V_BD_* enhances PFOM, higher *R_ON-SP_* can offset them. Illustrated in Figure 12 is Baliga’s figure of merit benchmark plot, which assesses various vertical GaN and SiC devices designed for *V_BD_* of approximately 900 V and above [36,41,59,66,67,69,71,72,73,115,116,117,118,119,120,121,122,123,124,125,126,127,128,129,130,131,132]. The plot suggests that, at present, GaN vertical devices lie mainly near the SiC limit, indicating the potential for further enhancements in GaN vertical technology to yield more efficient power devices.

## 6. Conclusions

Vertical GaN MOSFETs are pivotal in the realm of power electronics due to their ability to provide higher efficiency, higher power density, fast switching ability, and good thermal performance. Consequently, these vertical GaN MOSFETs find extensive utility across a broad spectrum of applications, including power supplies, electric vehicles, renewable energy systems, and data centers, among others. This article delves comprehensively into the accomplishments achieved thus far in the realm of GaN vertical devices. Additionally, it offers a thorough examination of various methodologies aimed at enhancing the breakdown voltage of these devices. The breakdown voltage is a critical parameter for power devices, as it governs their voltage-handling capacity, overvoltage protection, power conversion efficiency, design adaptability, and overall system reliability. Furthermore, while many of the techniques for improving the breakdown voltage of GaN vertical devices are consistent, it is worth noting that, in the case of GaN vertical MOSFETs, factors such as gate oxide thickness and quality also play a pivotal role in augmenting the breakdown voltage of these devices.

## Figures and Tables

**Figure 1 micromachines-14-01937-f001:**
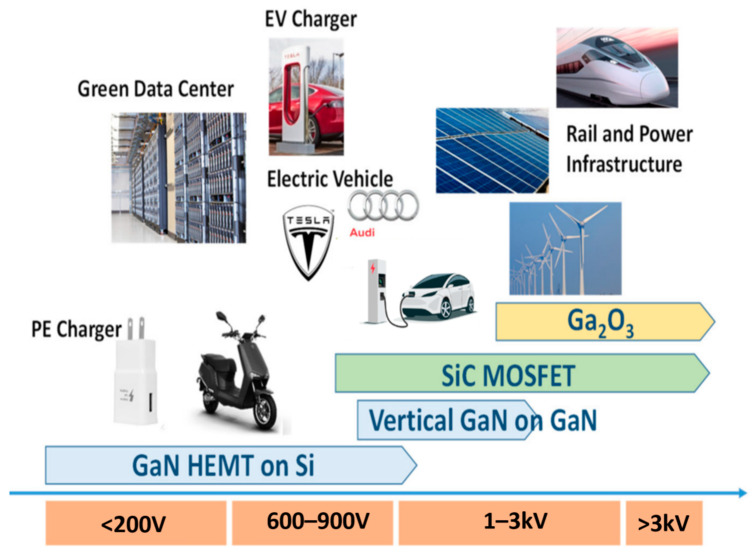
The distribution of wide bandgap power devices is based on voltage range [19].

**Figure 2 micromachines-14-01937-f002:**
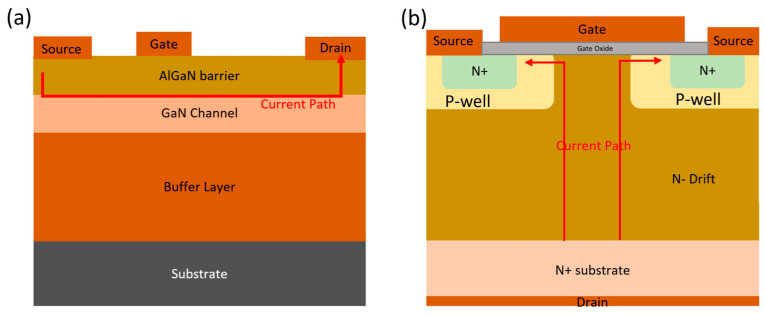
The schematic structures of (**a**) a lateral GaN device and (**b**) a vertical GaN device.

**Figure 3 micromachines-14-01937-f003:**
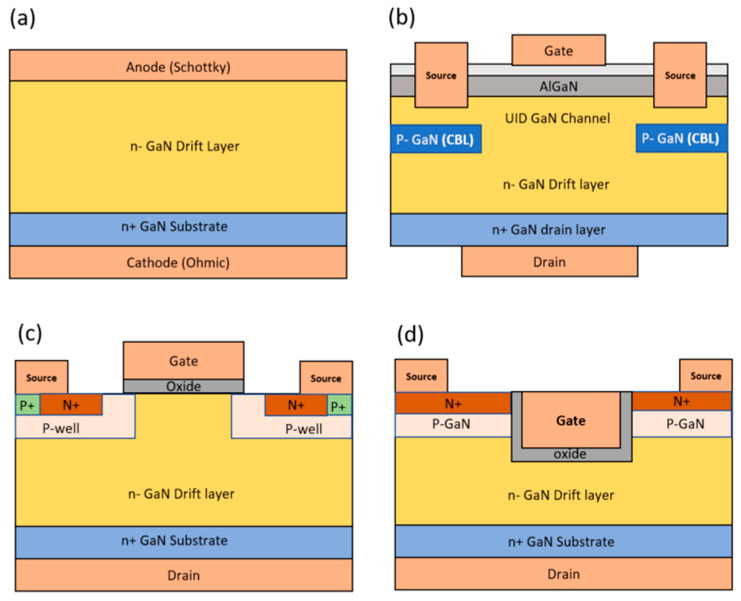
Vertical GaN power devices include (**a**) GaN power rectifier, (**b**) GaN CAVET, (**c**) GaN planar MOSFET, and (**d**) GaN Trench MOSFET.

**Figure 4 micromachines-14-01937-f004:**
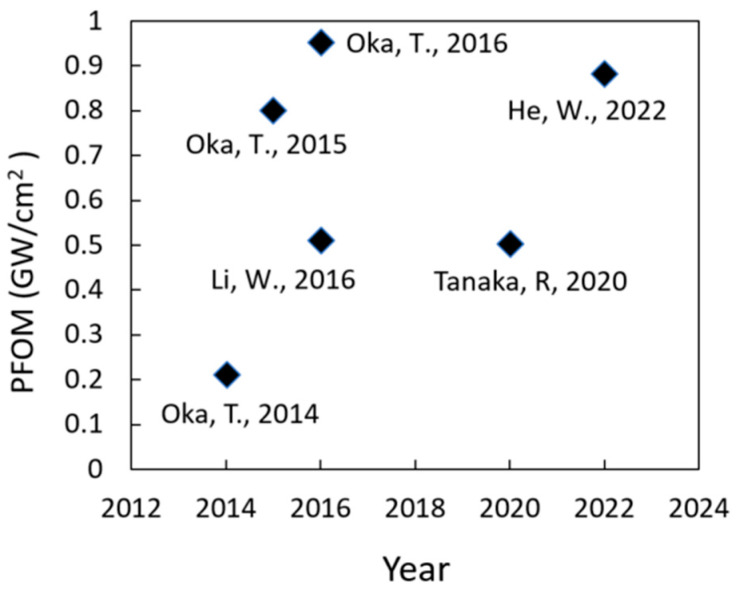
The comparison of PFOM for various vertical GaN MOSFET with a breakdown voltage of 1 kV and above [41,59,66,67,68,69].

**Figure 5 micromachines-14-01937-f005:**
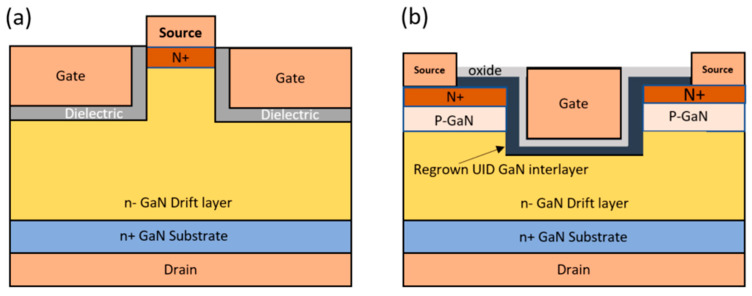
(**a**) Fin structure vertical GaN MOSFET and (**b**) in situ oxide–GaN interlayer FET (OG_FET).

**Figure 6 micromachines-14-01937-f006:**
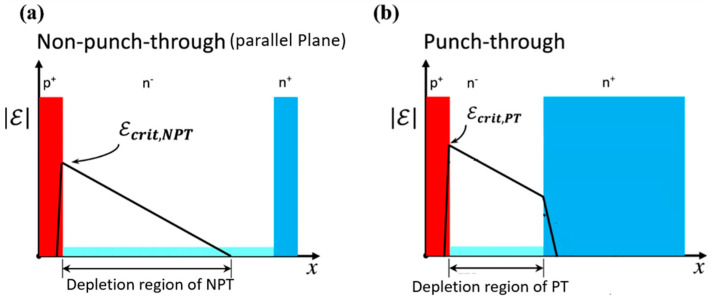
The schematic representation of the electric field distribution profiles for two distinct breakdown scenarios: (**a**) Non-punch-through (parallel-plane) breakdown, and (**b**) punch-through breakdown [88].

**Figure 7 micromachines-14-01937-f007:**
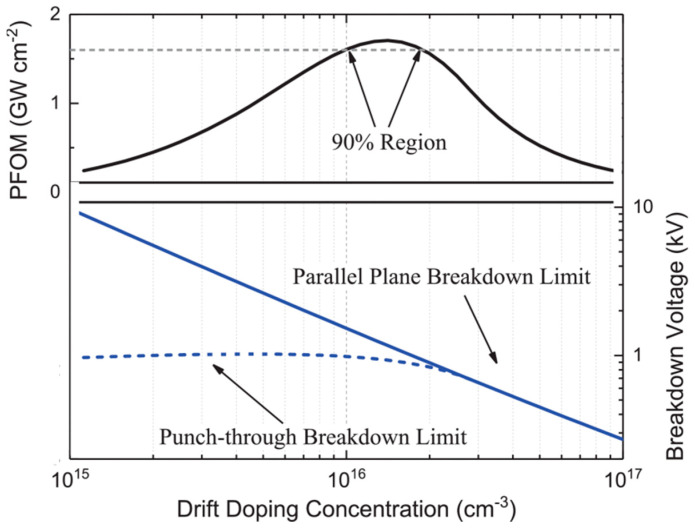
(**Top**) PFOM vs. doping concentration, and (**bottom**) breakdown vs. doping concentration. Reprinted with permission from Ref. [85]. 2023, E. Brusaterra.

**Figure 8 micromachines-14-01937-f008:**
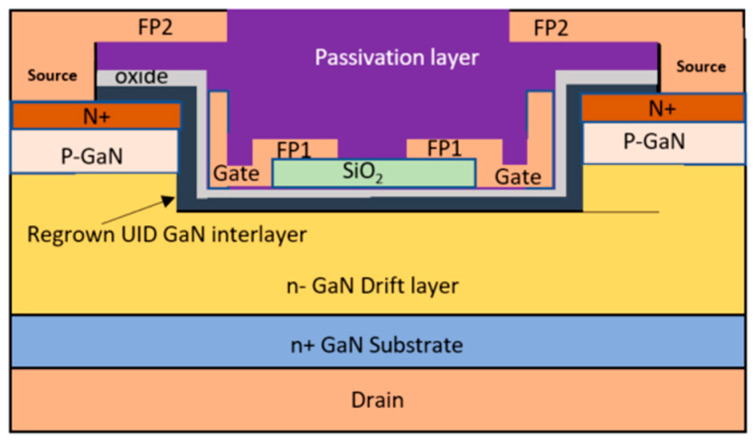
GaN MOSFET with double field plate Reprinted with permission from Ref. [73]. 2017, Dong Ji.

**Figure 9 micromachines-14-01937-f009:**
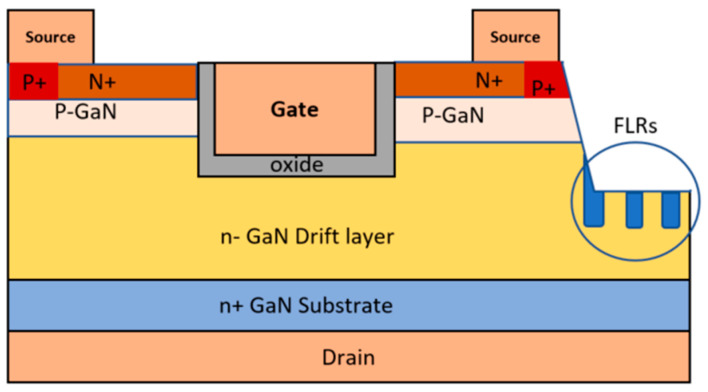
GaN MOSFET with field-limiting rings (FLRs). Reprinted with permission from Ref. [106]. 2022, Kachi Tetsu and Narita Tetsuo.

**Figure 10 micromachines-14-01937-f010:**
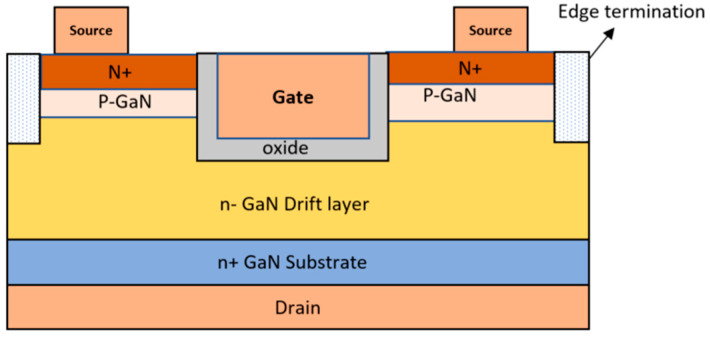
Mesa edge termination of GaN MOSFET. Reprinted with permission from Ref. [96]. 2019, Srabanti Chowdhury.

**Figure 11 micromachines-14-01937-f011:**
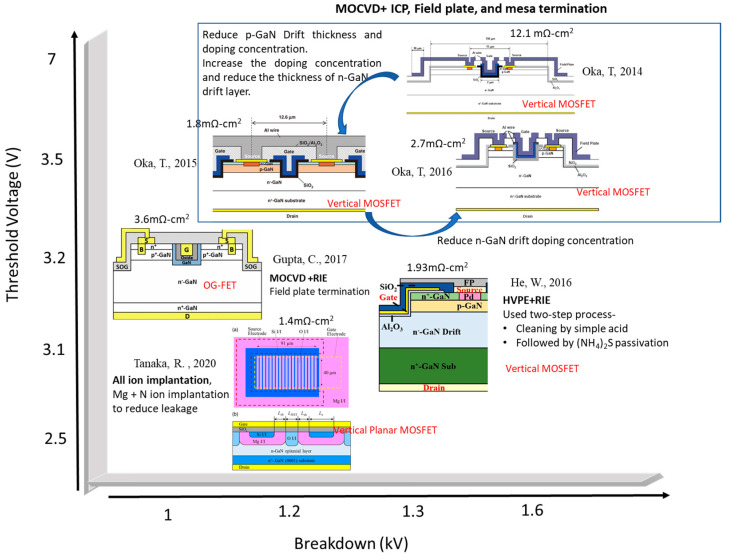
The schematic comparison of GaN vertical MOSFET with a breakdown voltage greater than 1 kV. Reprinted with permission from Refs. [41,59,66,67,69,115].

**Figure 12 micromachines-14-01937-f012:**
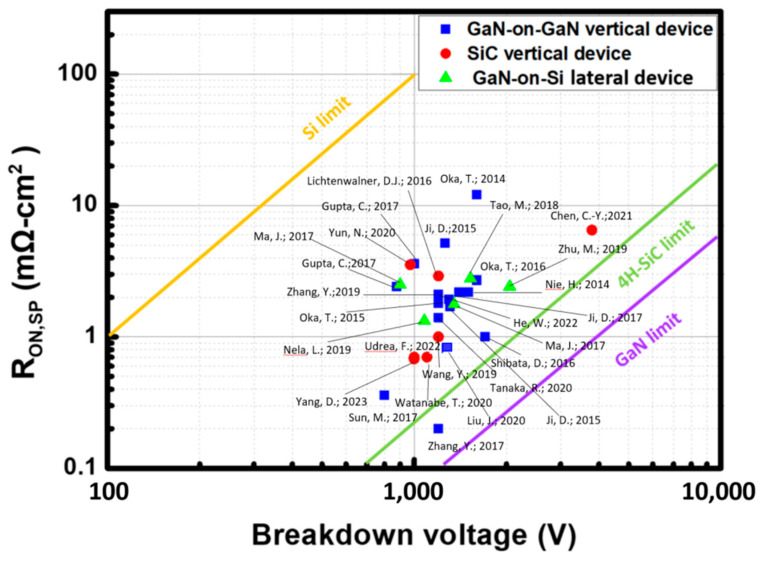
Benchmarking vertical GaN devices against lateral GaN MOSFET devices and vertical SiC devices. Adapted from Refs. [36,41,59,66,67,69,71,72,73,115,116,117,118,119,120,121,122,123,124,125,126,127,128,129,130,131,132].

**Table 1 micromachines-14-01937-t001:** The comparison of various vertical GaN transistors with a breakdown voltage greater than 1 kV.

Year	Type	Substrate	Threshold Voltage (V)	Breakdown Voltage (kV)	Specific On-Resistance (mΩ-cm^2^)	PFOM	References
2014	CAVET	GaN	0.5	1.5	2.2	1.022	[116]
2014	Vertical trench MOSFETVert	GaN	7	1.6	12.1	0.211	[41]
2015	Vertical trench MOSFETVer	GaN	3.5	1.2	1.8	0.8	[66]
2016	CAVET	GaN	2.5	1.7	1.0	2.89	[131]
2016	Vertical trench MOSFET.	GaN	3.5	1.6	2.7	0.95	[67]
2017	Vertical trench MOSFET (OG-FET)	GaN	3.2	1	3.6	0.28	[115]
2017	GaN OG-FET	GaN	4.7	1.4	2.2	0.89	[73]
2017	GaN fin FET	GaN	1	1.2	0.2	7.2	[72]
2019	Vertical Power FinFETs	GaN	1.3	1.2	2.1	0.68	[119]
2020	Vertical GaN planar MOSFET	GaN	2.5	1.2	1.4	1.03	[59]
2022	Vertical trench MOSFET	GaN	3.1	1.3	1.93	0.87	[69]

## Data Availability

Data sharing not applicable.

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
