# Peer review of "Vertical GaN MOSFET Power Devices"

_micromachines, 2023, doi:10.3390/mi14101937_

Round 1

Reviewer 1 Report

The manuscript is a well-structured and informative review. It emphasizes the significance of GaN power devices, with a specific focus on vertical GaN MOSFETs. The clear organization and explanations enhance its readability.

The section on methods to improve the breakdown voltage is noteworthy, providing insights into drift layer doping, field plate termination, and edge termination. 

The comments are as following. 1. The abbreviations of variables such as VBD, Ron, VDS,max, etc., should be in italics. 2. L10 ~ L44, regarding the statement "The maximum drain ... 2 mOhm to 200 mOhm," please provide references to support this conclusion. Furthermore, it is advisable to avoid using vague terms like "might" or "about" in scientific literature when describing data, similar terms are also used in line 50. 3. L464 ~ L467, what is the meaning of references 7 to 10? Are these references from commercial device datasheets? For a review, the accuracy of references is crucial; please list the corresponding reference URLs. 4. The formatting of the references is inconsistent, and some references contain unnecessary DOI URLs. Please make corrections. 5. L58 ~ L60, "However, ... SiC counterpart." Here, you referenced the work of other research groups, but why wasn't it cited? 6. L158 ~ 162, there is a contradiction between the two conclusions, "reduced cost" and "high fabrication cost." While I understand your point, it would be better to rephrase the language. Additionally, the reference [43] cited here is from 2017. When discussing the current research status, it's preferable to cite more recent literature. Overall, there is a limited number of references from the past three years in this paper. 7. L163, "3. Vertical GaN tranistors Devices" provides an overview of the progress in vertical GaN transistor devices. It would be beneficial to include a graph here. Use the publication year as the x-axis and the PFOM of the devices as the y-axis to enhance visibility. 8. Line 254, "4. Methods To improve the Breakdown Voltage" mentions three common techniques to enhance breakdown voltage in devices but does not provide a comprehensive comparison of their pros and cons. Please discuss which technique holds more promise and why. This is an important consideration when writing a review.

Reviewer 2 Report

Nice review paper. However, it does not discusses a very important field in vertical GaN device research. That is GaN on SiC. There are multiple papers on it. Please add a new section. Discuss GaN on SiC. Its benefit, disadvantages. Also, a quick search on the google scholar shows a lot of interesting research on Vertical GaN after 2021. Your review misses most of them. Please add as many of them as possible. To be a review paper you need to reflect the latest on the field. 

Reviewer 3 Report

The authors present a review of vertical GaN MOSFET power devices which possess some notable advantages, including heightened voltage capability, reduced on-resistance, and enhanced switching speeds. However, it is evident that this review suffers from organizational shortcomings as it predominantly centers on the device's architectural aspects, lacking a comparative analysis of device performance through the utilization of tables. Furthermore, numerous syntax errors and formatting issues detract from the manuscript's overall quality. Consequently, I recommend that this submission be rejected.

1>   There are several grammatical issues present in the manuscript. For example, the term "Vertical GaN" in the abstract, the caption of Figure 2 which reads, "Schematic structures of (a) Lateral and (b) Vertical GaN power devices," and the heading "3. Vertical GaN Transistors Devices" on Page 4. Additionally, there is an error in Page 5 where "1.8 mΩ·cm2" is not formatted correctly. On Page 7, the section heading "Methods To Improve the Breakdown Voltage" requires proper formatting, and there is a misspelling in the subheading "4.1. Optimizing Drift Layer Doping Concentration and Thickness." Lastly, the caption for "Figure 7. GaN MOSFET with Double Field Plate" contains an error in the word "double."

2>   The figure captions exhibit confusion in their numbering. For instance, the caption for "Figure 4. Fin Structure and In-situ Oxide-GaN Interlayer FET (OG_FET)" lacks corresponding subfigures labeled as 4a and 4b.

3>   The references section exhibits numerous formatting issues, including missing journal names for references (Ref. 4, 5, 6, etc.), inconsistent capitalization of titles (e.g., Ref. 1 - "Smart Power Devices and ICs Using GaAs and Wide and Extreme Bandgap Semiconductors," Ref. 6 - "GaN-Based Power Devices: Physics, Reliability, and Perspectives"), and an error in the formatting of "Ref. 25 - Physica Status Solidi (A)," among others.

4>   The manuscript requires improvement in English language proficiency. One instance where this is evident is in the sentence: "In addition, the different methodologies to improve the breakdown voltage of the device is reviewed.".

The authors present a review of vertical GaN MOSFET power devices which possess some notable advantages, including heightened voltage capability, reduced on-resistance, and enhanced switching speeds. However, it is evident that this review suffers from organizational shortcomings as it predominantly centers on the device's architectural aspects, lacking a comparative analysis of device performance through the utilization of tables. Furthermore, numerous syntax errors and formatting issues detract from the manuscript's overall quality. Consequently, I recommend that this submission be rejected.

1>   There are several grammatical issues present in the manuscript. For example, the term "Vertical GaN" in the abstract, the caption of Figure 2 which reads, "Schematic structures of (a) Lateral and (b) Vertical GaN power devices," and the heading "3. Vertical GaN Transistors Devices" on Page 4. Additionally, there is an error in Page 5 where "1.8 mΩ·cm2" is not formatted correctly. On Page 7, the section heading "Methods To Improve the Breakdown Voltage" requires proper formatting, and there is a misspelling in the subheading "4.1. Optimizing Drift Layer Doping Concentration and Thickness." Lastly, the caption for "Figure 7. GaN MOSFET with Double Field Plate" contains an error in the word "double."

2>   The figure captions exhibit confusion in their numbering. For instance, the caption for "Figure 4. Fin Structure and In-situ Oxide-GaN Interlayer FET (OG_FET)" lacks corresponding subfigures labeled as 4a and 4b.

3>   The references section exhibits numerous formatting issues, including missing journal names for references (Ref. 4, 5, 6, etc.), inconsistent capitalization of titles (e.g., Ref. 1 - "Smart Power Devices and ICs Using GaAs and Wide and Extreme Bandgap Semiconductors," Ref. 6 - "GaN-Based Power Devices: Physics, Reliability, and Perspectives"), and an error in the formatting of "Ref. 25 - Physica Status Solidi (A)," among others.

4>   The manuscript requires improvement in English language proficiency. One instance where this is evident is in the sentence: "In addition, the different methodologies to improve the breakdown voltage of the device is reviewed.".

Round 2

Reviewer 2 Report

The paper can be accepted in present format. 

Reviewer 3 Report

It can not be accpeted  in current form. There are still numerous syntax errors and formatting issues in manuscript.

1. Table 1, the year of "20172020" shouls be 2017 or 2020. Additionally, ":- A low-damage RIE dry etch was used to obtain near-vertical sidewalls (10º from vertical)." and "Clbased gases using inductively coupled plasma (ICP)." can be removed from the table.

2. Some references have publication dates, some do not. Ref. 55 and 59, "2018/06/12 2018" should be "2018"; "2020/01/27" changes to "2020". 

OK
